# Spatiotemporal Feature Enhancement Aids the Driving Intention Inference of Intelligent Vehicles

**DOI:** 10.3390/ijerph191811819

**Published:** 2022-09-19

**Authors:** Huiqin Chen, Hailong Chen, Hao Liu, Xiexing Feng

**Affiliations:** 1College of Mechanical Engineering, Hangzhou Dianzi University, Hangzhou 310018, China; 2Department of Electrical and Computer Engineering, University of Windsor, Windsor, ON N9B 2T6, Canada

**Keywords:** driving intention inference, two-stream networks, spatiotemporal features, intelligent vehicle

## Abstract

In order that fully self-driving vehicles can be realized, it is believed that systems where the driver shares control and authority with the intelligent vehicle offer the most effective solution. An understanding of driving intention is the key to building a collaborative autonomous driving system. In this study, the proposed method incorporates the spatiotemporal features of driver behavior and forward-facing traffic scenes through a feature extraction module; the joint representation was input into an inference module for obtaining driver intentions. The feature extraction module was a two-stream structure that was designed based on a deep three-dimensional convolutional neural network. To accommodate the differences in video data inside and outside the cab, the two-stream network consists of a slow pathway that processes the driver behavior data with low frame rates, along with a fast pathway that processes traffic scene data with high frame rates. Then, a gated recurrent unit, based on a recurrent neural network, and a fully connected layer constitute an intent inference module to estimate the driver’s lane-change and turning intentions. A public dataset, Brain4Cars, was used to validate the proposed method. The results showed that compared with modeling using the data related to driver behaviors, the ability of intention inference is significantly improved after integrating traffic scene information. The overall accuracy of the intention inference of five intents was 84.92% at a time of 1 s prior to the maneuver, indicating that making full use of traffic scene information was an effective way to improve inference performance.

## 1. Introduction

Studies published in the field of road safety have shown that most traffic accidents are caused by inappropriate behavior, such as driver operation errors [1]. Correctly understanding the driver’s intention is considered to be an effective way to reduce traffic injuries and improve traffic safety [2]. With improvements in intelligent driving technology, it has become a consensus that the richness of experience of human drivers in complex driving tasks can be used as a guide for automated vehicles [3,4,5]. Compared with a single driving mode, human–machine collaborative driving has higher safety and efficiency and has become a potential solution for improving driving safety. However, the mutual understanding between the driver and the automated system is a prerequisite for human–machine interaction, which helps the automated vehicle to make decisions in uncertain environments that are more acceptable to the driver [6,7]. Driving intention is a driver’s response to dynamic changes in nearby traffic situations, and the intention inference is that the driving system predicts the driver’s purpose according to the sequence of time. From a cognitive psychology perspective, intentions are difficult to predict, as invisible thoughts that precede actual actions [8]. The continuous enhancement of the comprehensive perception of intelligent vehicles can make the inference of driving intention traceable. For example, information such as the driver’s physical behavior and the traffic environment can be implemented for short-term driving intention inference. If the agent of an intelligent vehicle can learn internal correlations by observing a series of historical data, the accuracy of intention inference can be further improved, thus providing more reliable support for the development of human–machine co-driving technology.

As explored in earlier research, machine learning techniques have been widely used in the field of intention recognition and prediction [9,10,11]. Jang et al. [9] proposed a driving intent classification method based on eye-movement analysis using a support vector machine (SVM). Amsalu et al. [10] proposed a driving intention classification model for intersection scenes, based on the hidden Markov model (HMM). Artificial neural networks (ANNs) can deal with the complex and implicit nonlinear relationships between variables [12], and they can also be a powerful tool for driving intention prediction when the datasets are large enough [13,14]. Kim et al. [13] used the artificial neural network model as the driver’s lane change intention preprocessing algorithm to enhance the information processing ability of the vehicle state and road conditions, thereby improving the driving intention prediction performance. However, ANNs usually focus on the independence of input data and do not apply to observation sequences, while time series are very important for driving intention references.

In recent years, the development of deep learning has provided new technical means for solving this problem. Long short-term memory (LSTM) has a strong context-modeling ability and is widely used to map sensor observation sequences to predict driving intention categories. Girma et al. [15] modeled vehicle speed and yaw rate information based on a deep bidirectional LSTM and predicted in advance whether the driver would go straight ahead, stop, turn right, or turn left. Tang et al. [16] proposed a multi-LSTM method that took the historical trajectory and speed of the vehicle as input and realized the classification of the vehicle’s lane-changing intention. Convolutional neural networks (CNNs) work similarly to the human visual system and can be used to extract and learn high-level representations of drivers’ physical behavior-related features; they are often used to identify drivers’ drowsiness [17] and distraction [18]. It has also been used to infer driving intention in some recent studies. Specifically, the CNN-RNN network is considered to be a powerful model for sequential image data processing [19]. In [20], a CNN was used as an encoder to extract the spatial features of driver behavior sequences, after which LSTM was used as a decoder for temporal modeling to infer driver intention. A three-dimensional convolutional neural network (3DCNN) is an extension of a two-dimensional convolutional neural network (2DCNN) to the temporal dimension, which can be used to directly extract the spatiotemporal features of image sequences [21], at which point driving intent prediction based on 3D convolutional networks has been applied. This kind of method can usually achieve end-to-end training without the need for handcrafted feature selection and extraction based on expert knowledge, which is beneficial for reducing the workload and time costs during the experiment. In a previous study [22], a vision-based 3D convolutional residual learning method was proposed that utilized optical-flow images in the cab to achieve driving intent prediction, but the calculation of optical flow often complicated the data processing. In addition, the modeling of the influence of surrounding vehicles is also worth considering. For example, Huang et al. [23] proposed a ConvLSTM-based vehicle intent prediction method for urban road scenarios. The method captured and modeled the interactions of surrounding vehicles from a bird’s-eye viewpoint, achieving approximately 60% accuracy. Taking the driver factor into consideration may further improve the model’s ability to discriminate intent.

Gebert [22], Xing [24], and others conducted a statistical analysis on the differences in behavior exhibited by different driving intentions and found that driving intentions had a high correlation with driver behavior. In addition, driver head posture and specific behavior, such as checking rear-view mirrors, can provide important evidence for intention inference. Although the driving intention inference method for the driver’s temporal behavioral dynamics model has achieved impressive results, there are different driving styles among various individuals; using driver behavior as the sole basis for intention inference tends to cause many uncertainties in the results. In addition, the driver may carry out a series of checking actions in the real world and then terminate the maneuver because of the currently risky environment. However, maneuvers may also occur without environmental checks, due to the driver’s high confidence. In these cases, the vehicle heading angle and steering wheel angle can provide complementary clues for intent inferences [25]. With the development of perception technology, it has become possible to obtain these clues through visual sensors. For example, Chen and Fernandez [26,27] used the front camera’s data stream to predict the steering angle.

In this study, driver behavior data comprise the main input of the proposed method, and the forward-facing traffic scene data stream is processed as an enhancement to the in-cab driver behavior data. The joint representation of these two data sources is the final basis for potential intention inferences. The contributions of this study can be summarized as follows: (1) based on the proposed method, it can be observed that using a low frame-rate slow pathway and a high frame-rate fast pathway to separately process the video streams inside and outside the cab would obtain stronger joint representation, thereby improving the driving intent inference ability; (2) the developed parallel two-stream structure can make full use of traffic scene information and facilitate feature fusion, supporting end-to-end learning using raw data; (3) the model has better adaptability. In addition, five intentions can be inferred with relative accuracy when either driver behavior data or traffic scene data are unavailable.

## 2. Methods

The spatiotemporal enhancement driving intention inference (STEDII) process can be viewed as a solution to the classification problem of sequence images. In general, for two-stream network-based action classification methods, one branch learns the spatial semantic information in the video, while the other branch extracts the temporal features from the video [28]. In our work, a novel driving intention inference framework, STEDII, was proposed that simultaneously processes the video data inside and outside the cab. As shown in Figure 1, the data layer is used to synchronize the two segments of video data and distribute them to the backbone, which is a parallel two-stream network. The basic structure of the backbone network is the SlowFast network [29]. In this module, the low frame rate channel, slow pathway (SP), is used to extract driver behavior features, and another similar high frame-rate channel fast pathway (FP) is used to process forward-facing traffic scene information. At each stage, the features extracted by the FP are incorporated into the SP for the purpose of enhancement. Finally, the joint representation is input to the inference module and the most likely intent is identified through the inference module. Each module is described in detail as follows.

### 2.1. Data Layer

In this section, to simplify the data preprocessing procedure and synchronize the internal and external video streams, the original video frames were rearranged to obtain mixed frames before entering the two-stream networks. The arrangement of the mixed frame sequence was determined by the ratio of the FP and SP frame rates, that is, the ratio of the number of images processed at the same time, which is denoted as α. For the convenience of designing the network structure, the values of α were taken as 1, 3, and 7 in the study. Figure 2 shows the case studies when α was taken as 1 and 3.

### 2.2. Spatiotemporal Feature Extraction, Based on Two-Stream Networks

As mentioned earlier, both branches of the feature extraction module have the ability to capture spatiotemporal features. To meet the needs of updating fast external video streams, FP has a higher temporal resolution. Therefore, there are dimensional differences in the features extracted by the two paths. Before incorporating the features extracted by FP into SP, the dimension-matching process needs to be carried out through the interactive connection, which will be introduced in detail in Section 2.2.3. The structures of SP and FP are introduced below.

#### 2.2.1. Slow Pathway

The structure of SP is an extension of ResNet [30] in the time dimension. An important feature of ResNet is the innovative residual units in the structure. This skip connection structure allows the signal to propagate directly from the last layer to any previous layer. The main advantage of gradient backward propagation is that intermediate weight layers that may cause gradient deterioration can be skipped, allowing the network to be built deeper. Through residual learning, the weights of multiple nonlinear layers can tend to move to zero when necessary, that is, to achieve identity mapping: y = x. Therefore, the learning objective of the stacked nonlinear layers becomes a new mapping of f(x): y − x = 0. To adapt to the feature extraction process, the residual blocks are designed in two forms: convolution blocks, with different output and input dimensions, and identity blocks with the same output and input dimensions, as shown in Figure 3a,b.

The internal driver behavior feature, *x*, is first input into stage 1, defined as Equation (1), which consists of a three-dimensional convolution layer (CONV3D), a BatchNorm3d layer (BN3D) [31], a ReLU [32] activation function, and a Maxpool3d layer. Here, *c* is the number of output channels of the convolution operation, *k* is the spatiotemporal kernel size with a temporal step set to 1 in SP, and *s* is the stride of the convolution operation.
(1)Stage1=Maxpool3d(ReLU(BN3D(CONV3D(x, c, k, s))))

Figure 3a is the convolution block (Conv_block), which is defined as Equation (2) [30], where $x_{l}$ and $x_{l+1}$ are the input and the output of the *l*th layer in the Conv_block. *f* is represented by ReLU, and ℱ is a nonlinear residual mapping model represented by convolutional kernel weights, $w_{l}$. For stage 2 and stage 3, the spatiotemporal kernel sizes were [(1 × 1 × 1), (1 × 3 × 3), (1 × 1 × 1)], respectively, of the three CONV3Ds in SP, but at stage 4 and stage 5, they were designed to be [(3 × 1 × 1), (1 × 3 × 3), (1 × 1 × 1)]. In FP, the kernel sizes of stage 2–stage 5 were [(3 × 1 × 1), (1 × 3 × 3), (1 × 1 × 1)].
(2)$$x_{l+1}=f\left(x_{l}+ℱ\left(x_{l},\;w_{l}\right)\right)$$

Figure 3b is the identity block described in Equation (3) [30]. A square matrix, $w_{s}$, is only used to match the dimensions. The input and output of the shortcut branch are equal, and the spatiotemporal kernel sizes of stage 2–stage 5 are [(3 × 1 × 1), (1 × 3 × 3), (1 × 1 × 1)].
(3)$$x_{l+1}=f\left(w_{s}x_{l}+ℱ\left(x_{l},\;w_{l}\right)\right)$$

The SP in Figure 1 was designed based on the convolution block and the identity block, the details of which are shown in Table 1. The output shape represents the height and width of the output feature of each layer, and the time dimension is the number of internal video frames, *N*.

#### 2.2.2. Fast Pathway

Parallel to SP, FP has a similar structure, and the details are shown in Table 2. However, unlike SP, there are some properties, elaborated on as follows [29]:High frame rate. According to the data layer, the number of images allocated to FP is α times that of SP, which also means that FP processes traffic scene video data at a higher speed;High temporal resolution. To ensure a high processing speed, the temporal dimension of the convolution kernel is greater than 1 throughout all stages. Therefore, the FP has the ability of detailed motion feature extraction while maintaining temporal fidelity;Few-channel capacity. In the study, the number of input channels of FP is 1/8 of SP, denoted as β = 1/8. Therefore, compared with SP, FP focuses more on temporal modeling and weakens the extraction ability of spatial semantic information.

#### 2.2.3. Spatiotemporal Feature Enhancement

As mentioned above, the motion features extracted from the forward-facing traffic scene video are used to enhance the driver behavior features, to improve the driving intention inference ability of the model. This idea is realized by the operation of fusing the features of the FP into the SP. Because the temporal dimensions of the two pathways are inconsistent, the interactive connection (shown in Figure 1) was implemented to match them. A sketch map is provided in Figure 4, which shows the feature match and the fusion process between the two streams.

As shown in Figure 4, the feature shape of the SP is denoted as {*N*, *S*^2^, *C*}, the feature shape of the FP is {α*N*, *S*^2^, *βC*}, where *S*^2^, represents the width × height, and *C* is the number of the channels. First, the feature shape of the FP is reshaped to {*N*, *S*^2^, 2*βC*} using a 3D convolution operation with 2*βC* output channels, and the stride is set to α. Second, the output feature of the interactive connection was fused into the SP by a concatenation operator in the channel direction, then the feature shape is adjusted to {*N*, *S*^2^, 2*βC* + *C*}.

In the last stage, the outputs of the two pathways are input into a global average pooling layer; that is, an average value is calculated for the features of each channel. At this time, the time dimension and height and width are set to [α*N*, 1, 1]. Therefore, the shapes of the output features of SP and FP are {α*N*, 1^2^, 2048} and {α*N*, 1^2^, 256}, respectively. After these two features are concatenated in the channel direction, the new feature shape is {α*N*, 1^2^, 2304}. Finally, a dropout layer is added to suppress overfitting and drop neuron connections with a probability of 0.5 during training.

### 2.3. Driver Intention Inference Module

The features extracted by the backbone network are fed into the recognition module (see Figure 5) for driving intent classification. The recognition module consists of GRU [33], a full-connection (FC) layer, and the softmax function. After the raw video frames are passed through the backbone network, an input sequence, *X* containing α*N* dimensions of 2304 can be obtained, where *X*: {α*N*, 2304}, *X* = (*X*_1_, *X*_2_, …, *X_t_*, …, *X*_α*N*_). First, *X* is input into a GRU network with one layer and 24 units. The structure of a gated cell unit is displayed in Figure 6. For a given input *X_t_* at time t, the hidden state *H*_*t*−1_ is at the previous time step. Then, the reset gate *R_t_*, the update gate *Z_t_*, and the hidden state *H_t_* are calculated using Equations (4), (5), and (7). The candidate hidden state, $\tilde{H_{t}}$, is calculated using Equation (6), as follows:(4)$$R_{t}=\sigma \left(X_{t}W_{xr}+H_{t-1}W_{hr}+b_{r}\right)$$
(5)$$\;\;Z_{t}=\sigma \left(X_{t}W_{xz}+H_{t-1}W_{hz}+b_{z}\right)$$
(6)$$\tilde{H_{t}}=\tanh\left(X_{t}W_{xh}+\left(R_{t}⊙H_{t-1}\right)W_{hh}+b_{h}\right)$$
(7)$$H_{t}=Z_{t}⊙H_{t-1}+\left(1-Z_{t}\right)⊙\;\tilde{H_{t}\;}$$
where $W_{xr}$, $W_{xz}$, $W_{xh}$ and $W_{hr}$, $W_{hz}$, $W_{hh}$ are weight parameters. $b_{r}$, $b_{z}$ and $b_{h}$ are bias parameters. *σ* and tanh represent the sigmoid function and the hyperbolic tangent function, respectively. ⊙ denotes an element-wise dot product.

Then, the features that are output by GRUs are mapped into 5 label spaces of video data by an FC layer, and the number of neurons in the output layer is set to 5. Finally, the class probability distribution is normalized using the softmax function, as shown in Equation (8). Additionally, since the backbone has completed spatiotemporal modeling, the fused features can be directly classified by FC after flattening, so as to simplify the model.
(8)$$softmax\left(z_{j}\right)=\frac{e^{z_{j}}}{\sum_{c=1}^{C}e^{z_{c}}}$$
where $softmax\left(z_{j}\right)$ represents the value of *j*th after the C-dimension vector *z* is mapped by the softmax functions. *C* represents the number of classes.

### 2.4. Implementation Details and Evaluation Metrics

#### 2.4.1. Implementation Details

The training process of STEDII adopted the strategy of transfer-learning. The backbone feature extraction network used Knetics-400 [35] (a human behavior dataset) to initialize the backbone weights. Then, the whole framework, including GRU networks (STEDII-GRU), was trained. An SGD optimizer with a momentum of 0.9 and a weight decay of 0.001 was used to train the model. The initial learning rate was set to 0.01, and CosineAnnealing was applied for the learning-rate policy. The model was trained on a GPU Nvidia RTX 3090 for a total of 60 epochs. The Brain4cars dataset [36] recorded a video sequence before maneuvering. A total of 625 samples were collected, including 234 straight-ahead samples, 140 left-lane change samples, 58 left-turn samples, 142 right-lane change samples, and 51 right-turn samples. After the few samples that were out of synchronization were discarded, 80% of the remaining video sequences were used for training and 20% for testing.

#### 2.4.2. Evaluation Metrics

In this study, accuracy, F1 score, the confusion matrix, parameter quantity, and prediction time were used to measure the driver intention inference performance of the proposed model and other models. The accuracy rate (Acc) and F1 score were calculated using Equations (9) and (12) [37,38], as follows:(9)$$Acc=\frac{TP+TN}{TP+FP+TN+FN}$$
(10)$$Pr=\frac{TP}{TP+FP}$$
(11)$$Re=\frac{TP}{TP+FN}$$
(12)$$F1\;score=\frac{2Pr\cdot Re}{Pr+Re}$$

In the equations, *TP* indicates that both the true label and the predicted label are positive; *TN* indicates that the true label and the predicted label are both negative; *FP* indicates that the true label is a negative class, and the predicted label is a positive class; *FN* indicates that the true label is a positive class, and the predicted label is a negative class. *Pr* and *Re* refer to the precision and recall rate, which are calculated using Equations (10) and (11), respectively, and the *F*1 score is the harmonic mean.

## 3. Results

In this section, the appropriate frame rate ratio was determined, initially based on driving intention inference accuracy and F1 score. Second, the proposed model was compared with four action recognition algorithms with different data input sources. Finally, the proposed model was compared with several classical models, and the inference results of five driving intentions: lane-keeping, left-lane change, left turn, right-lane change, and right turn were obtained.

### 3.1. Comparison of the Performance on Different Frame Rate Ratio

As mentioned earlier, the fast and slow pathways had different frame rates. It was important to select a reasonable frame ratio of slow and fast pathways in the STEDII-GRU model. In the pre-experiment, the frame rate ratios of the slow pathway and the fast pathway were set as 1:7, 1:3, 1:1, and 3:1 for convenience, and the corresponding α values were 7, 3, 1, and 1/3. A total of five sets of comparative experiments were carried out, and each set of experiments used the video data of 1 s, 2 s, 3 s, 4 s, and 5 s prior to the maneuver as the input of the model. The accuracy rate (Acc) and the average F1 score were applied to present the trend of model performance with different frame rate ratios, and the frame rate ratio that made the Acc and F1 score the highest was selected. The results are shown in Figure 7.

Figure 7 shows that with the same input, different frame rate ratios had various effects on the accuracy and F1 score of the model. When the frame rate ratio was 1:3, the model performed best. Compared with 1:1, on average, the accuracy and F1 score increase by 2.62% and 4.16%, respectively. The processing of driver behavior data and forward-facing traffic scene data at a frame ratio of 1:3 was better than at other ratios. Therefore, the frame rate ratio of 1:3 was maintained in the formal experiment.

### 3.2. Driver Intention Inference

Many researchers applied action recognition approaches to driving intention prediction, such as 3DResNet and ConvLSTM+3DResNet, in Gebert and Rong’s studies [22,39]. Additionally, TimeSformer [40] and TANet [41] were popular methods for video action recognition algorithms. A comparison of STEDII-GRU with these methods was carried out, and the results are shown in Table 3. All models were pre-trained using the Kinetics-400 dataset [35], and all video data with a capacity of 5 s were used as input. Table 3 shows the results with three different inputs, which included only the inside driver behavior video (I-only), only the outside traffic scene video (O-only), and both inside and outside videos (in and out). The average accuracy and F1 score, based on fivefold cross-validation, were used to illuminate the performance of different methods, where “*S**D*” represented the standard deviation and “Param” represented the parameter amount of the whole model, reflecting the complexity of the model.

According to Table 3, all algorithms achieved good results when using only inside driver behavior data, but the accuracies were between 77% and 84%, which was difficult to improve upon. The supplementary use of forward-facing traffic-scene data was a good solution. The experiments showed that the accuracy of intent inference was improved by approximately 6.6% and 7.3% when the ConvLSTM+3DResNet and STEDII-GRU models used the in-and-out video as input. However, the credibility of the TANet method decreased with the in-and-out input form, which may be due to the conflicting descriptions of the inner and outer motions by its adaptive convolution kernel. Notably, with the input form of outside-only (O-only), STEDII–GRU achieved an accuracy rate of 84.75% and an F1 score of 85.30%, which illuminates the finding that the proposed method had a strong motion capture ability and can meet the requirement of only traffic scene input. To better illustrate the effect of spatiotemporal enhancement, the regions of interest of the STEDII-GRU model under various input modalities were visualized. Figure 8 shows the visualization of the sequence of images before the driver’s left-lane change.

As shown in Figure 8a, the driver’s head movement is an important inference basis in the inside-only (I-only) input form. Specifically, the model focuses on areas such as the lower jaw, where movement is more obvious. Under the O-only input form (in Figure 8b), road conditions, such as lane lines, are the areas of concern for the model. In the input form of in-and-out (in Figure 8c), head rotation is still an important feature of attention and tracking. In the case of the outside video, the model mainly captures the lateral movement of pixels. Due to the mirroring principle, the recorded direction of the driver’s head movement is opposite to the actual turning movement. A possible explanation of motion feature enhancement is that when the driver’s head moves in the same direction as the forward-facing traffic scene, the relative motion between them is exacerbated.

In this study, a comparison with other existing machine learning models was also carried out in terms of prediction accuracy. Several classical methods are described as follows: (1) the support vector machine (SVM). The SVM is regarded as a classifier that was used to classify the features extracted by pre-trained MobileNetv2 [42] in the study. Since SVM cannot directly process sequence data, the input features were flattened. (2) The hidden Markov model (HMM). HMM is a generative model that could distribute a probabilistic graph for each class [36]. (3) A bi-directional long short-term memory (Bi-LSTM). MobileNetv2 was also used to extract the features. To facilitate information transfer between adjacent video frames, a temporal shift module (TSM) [43] was introduced into this model. Similar to CNN-RNN-M [25], the Bi-LSTM was used to model the temporal series. (4) STEDII, fully connected (FC). STEDII-FC used only the fully connected layers for classification. (5) STEDII-flow (FL). The optical flow was extracted from the traffic scene data using FlowNet [44]. The input mixed frame consisted of RGB images and optical-flow images in this study. To evaluate the performance of all methods, the precision and recall of each maneuver were calculated, including lane-keeping, a left-lane change, a left turn, a right-lane change, and a right turn. The overall accuracy of each method was given. The comparison results are shown in Table 4, and the models were evaluated based on the data collected at 1 s before the maneuver.

According to Table 4, the STEDII-GRU achieved an accuracy of 84.92%, which showed a great improvement over other classical models. Since the backbone network already had temporal modeling capabilities, the STEDII-FC did not require a separate decoder, such as Bi-LSTM. It showed that using only the fully connected layers for classification could also yield good results. Furthermore, the results of the STEDII-FL test showed that the performance of the model was not improved when optical-flow images were used as input. Therefore, the proposed method did not rely on the motion features provided by optical-flow images. The application of raw RGB images as input simplified the model’s data processing.

The confusion matrices of the proposed three methods are displayed in Figure 9, demonstrating the classification performance for the five intentions. The results showed that monitoring lane-keeping intention achieved the most accurate results (reaching 93% accuracy) with all methods. At the same time, the ability to recognize the intention of changing lanes to the left was not high (approximately 70% accuracy), and was easily confused with going straight ahead and turning left. The following three reasons are suggested, through the analysis of the misclassified samples. First, for some lane-keeping behaviors, the driver may perform some left or right checking behaviors to ensure that they are driving safely, which makes it easy to infer this as a left-lane change or a right-lane change. Second, some left-lane change intentions are very similar to the driver’s behavior in left-turn intentions, which may confuse the model. Finally, in some complex traffic scenarios, the driver may simultaneously check the traffic conditions on both sides and adjust the driving direction repeatedly, which poses challenges for intent prediction.

To further illuminate the prediction performance of the STEDII-GRU, the prediction times until maneuvering were set backward every 0.5 s. In addition, the performance of different input data sources is displayed in Figure 10. The “only driver” and “only traffic” variables represent the input forms of the “I-only” and “O-only” fields, respectively. This reveals that the inference of using only the forward-facing traffic features was more accurate than using only the driver features in the right region of −1.0 s. As the prediction time node moved left from −1.0 s, the “only traffic” line was rapidly driven down, while the “only driver” line remained high. This indicates that traffic-related features play an important role in the later driving intention inference process. Meanwhile, the accuracy of the model is greatly improved by combining driver behavior and traffic features. With advances in the prediction time to maneuvering, the useful information in the forward-facing traffic scene is reduced and its enhancement effect on driver-related features is weakened. Driver behavior data play a vital role throughout the prediction phase.

## 4. Discussion

In this study, an approach using spatiotemporal feature enhancement to aid driving intention inference for intelligent vehicles is presented. The results showed that the overall accuracy of intention inference of five intents was 84.92% at the time of 1 s prior to the maneuver.

The proposed method, STEDII-GRU, could make full use of the traffic scene information and enhance the driver behavior features. Driving intentions could largely be expressed by driving behaviors [45]. However, when only relying on the driver’s physical behaviors as the basis for the inference of intention, the accuracy rate was at a relatively low level. Traffic scenarios as supplementary inputs were considered an effective way to improve the performance of the driving intention inference model. For example, the research published by Rong [39] et al. showed that the internal and external features were complementary, but the method did not fully consider their differences, and there was still room for improvement in terms of accuracy. In some cases, incorporating traffic scene data even reduced the accuracy [22]. Aiming at addressing the difference between the video data from inside and outside the cab, the frame rate ratio was applied for the first time. Specifically, a low frame-rate network was used to extract the spatiotemporal features of driver behavior, while another branch network with a high frame rate was used to extract the features of forward-facing traffic scenes. In the experiments, the best results were achieved when the frame rate ratio of the two pathways was 1:3, and the improvement in accuracy near the start of the maneuver was more obvious.

A parallel two-stream structure was developed for extracting the inside and outside video motion features, respectively. The proposed method did not need to introduce an additional lane detection algorithm to extract the contextual traffic information [46] and avoided collecting driver’s head–eye behavior features [47]. Optical flow is a common way to assist in extracting motion features, as in the study published by Lv et al. [48]. However, raw RGB video frames could meet the input requirements of the proposed model and the data processing could be simplified. In addition, a reduction in the number of channels compressed the spatial capacity. The fast pathway could offer the light and flexible processing of forward-facing video frames with less of a computational cost while achieving higher inference accuracy with a lower number of parameters. Moreover, the interactive connection enabled the features of the two branch pathways to be deeply fused at an early stage. The visualization of the feature maps shows that the driver’s head rotation was an important inference basis; the spatiotemporal features extracted from the forward-facing traffic scene can be used for further enhancement. Furthermore, the proposed method was easy to extend due to its two-stage structure. When more features are available, they can be regarded as input into GRUs in the inference module. When integrating different features, only the inference module needs to be updated, thereby avoiding the impact of sensor changes on the entire system [49].

The model had a strong robustness. Unlike the method employed by Xing et al. [25], which cleaned the samples of unexpected driver behavior, occlusion, and exposure in the original dataset, all the available samples were retained to improve the model’s environmental adaptability. In daily driving, the intention inference system will inevitably be affected by the above factors. In addition, instead of predicting lane-change intentions, the turning intention, which was easily confused with the left/right lane-change intention was also included in this study. It showed that inference for all five intents achieved high accuracy and also performed well on each intent. In some studies, digital maps have been used as input to predict driving intentions [36], but this approach may not be suitable for unmapped roads [24]. The input of the STEDII-GRU model was collected from the cameras that are commonly found in smart vehicles, and the intention inference task was accomplished in a fully visual way. Another problem that may be encountered during actual driving is that users’ driving behavior data cannot be obtained due to the requirements of privacy protection. In this case, only the forward-facing traffic scene can also be used as an input in the STEDII-GRU model. According to the experimental results, an accuracy of 79.28% can be achieved by the proposed method for short-term intention prediction, which predicted behavior 1 s before the maneuver.

There are some limitations to this study that should be investigated in its eventual application. For example, the intention inference with a longer advance time mainly relied on driver-related features. The driving behavior would change substantially if the driver did not focus on the driving task due to fatigue or distraction [49]. Therefore, the intention inference system, combined with an estimation of the driver’s state, is expected to be more robust. Furthermore, in addition to complex road conditions that may terminate the intended maneuver [50], the driving states of other road users may also have a large impact on driving intentions. Future work is necessary to improve our overall understanding of the environment by promoting interaction with surrounding traffic conditions, with the goal of increasing confidence in driver intention inferences. In addition, the dataset used in this study contains general driving scenarios and does not involve some specific scenarios. If there are more datasets available in the future, more driving intention inferences can be conducted.

## 5. Conclusions

In this study, an end-to-end driving intention inference framework, based on spatiotemporal enhancement, named STEDII-GRU, has been proposed. First, a pre-trained two-stream network, the SlowFast network, was used as the backbone for feature extraction, and the high and low frame rate pathways of the two-stream network were used to process the outside forward-facing traffic scenes and inside driver behavior video data, respectively. Then, the joint spatiotemporal features were fed into a GRU to obtain the most likely intent. The model was validated on Brain4Cars, which is a naturalistic dataset containing highway and urban road driving information. The results showed that the model maximized the use of traffic information when the frame ratio rate of slow and fast pathways was 1:3. Compared to other action-recognition algorithms and classical methods, our model achieved the best performance. All five intents can be inferred at 1 s prior to the maneuver, with an overall accuracy of 84.92%. For further study, more valid samples from a large number of drivers need to be collected to improve the work. In addition, more sensors would be accessed in the future to train the model with more features, to validate its extending capability.

## Figures and Tables

**Figure 1 ijerph-19-11819-f001:**
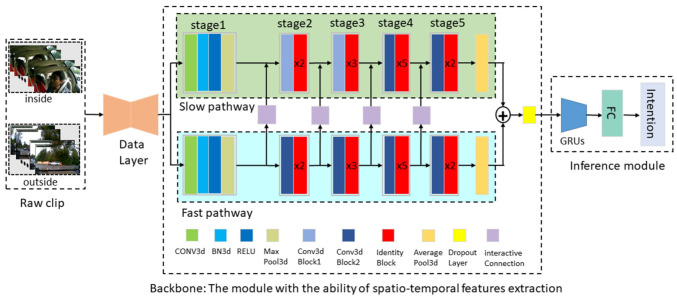
The framework of the STEDII model for driver intention inference.

**Figure 2 ijerph-19-11819-f002:**
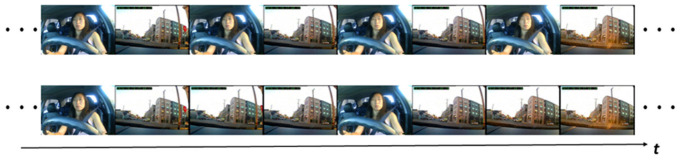
Recombined frame sequences. The upper and bottom graphs show the frame sequences of α = 1 and α = 3, respectively. The *t* represents the time axis.

**Figure 3 ijerph-19-11819-f003:**
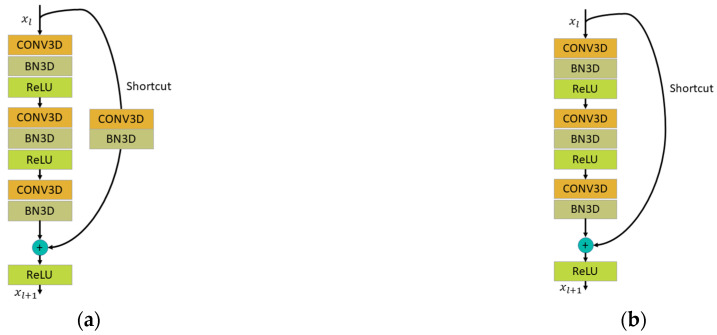
The residual blocks in the slow pathway in the forms of a convolution block and an identity block. (**a**) Convolution block. (**b**) Identity block.

**Figure 4 ijerph-19-11819-f004:**
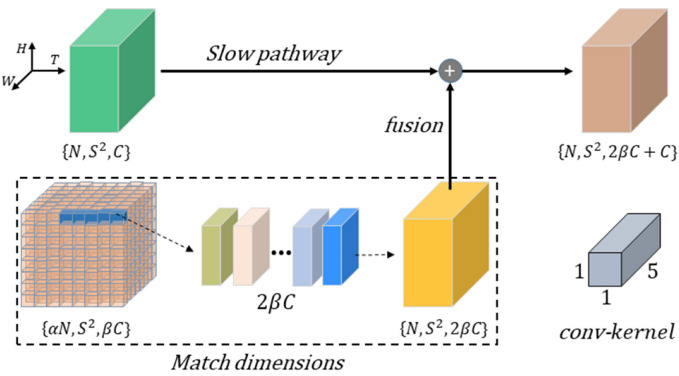
The feature match and fusion process of interactive connection.

**Figure 5 ijerph-19-11819-f005:**
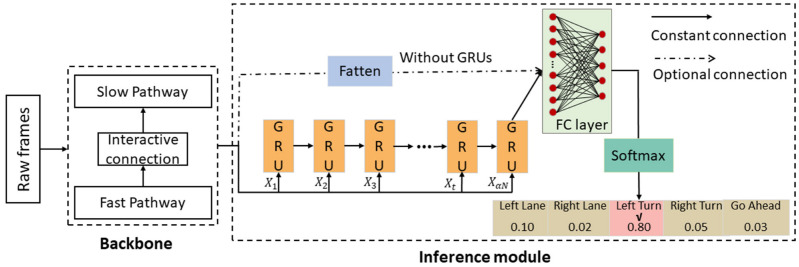
Driver intention inference module. The constant connection is the default. The optional connection represents a scenario where the fused features are flattened and sent directly to the fully connected layer without going through GRUs.

**Figure 6 ijerph-19-11819-f006:**
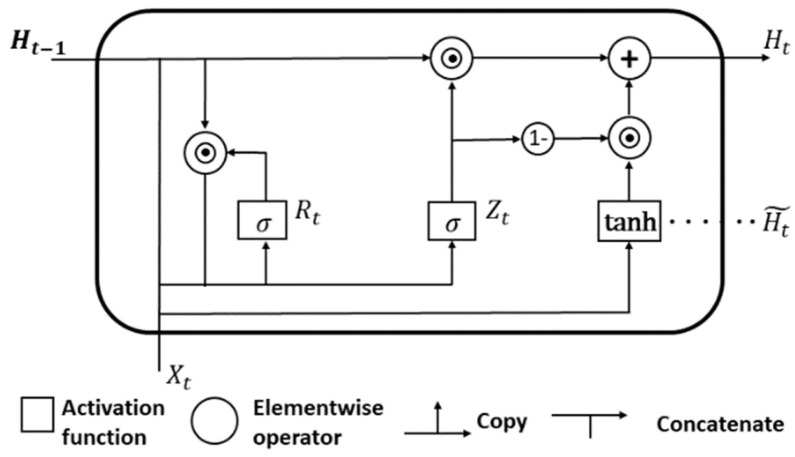
Structural diagram of GRU [34].

**Figure 7 ijerph-19-11819-f007:**
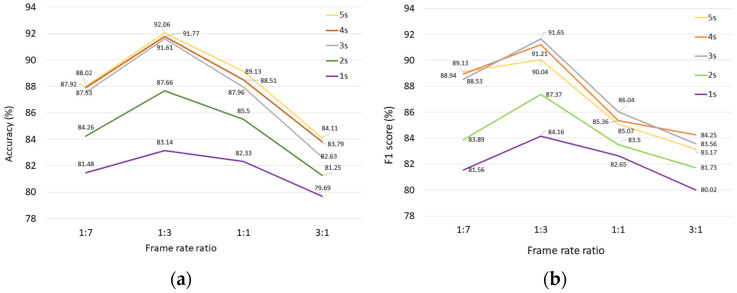
The trend of model performance with different frame-rate ratios. (**a**) Accuracy comparison results. (**b**) F1 score comparison results.

**Figure 8 ijerph-19-11819-f008:**
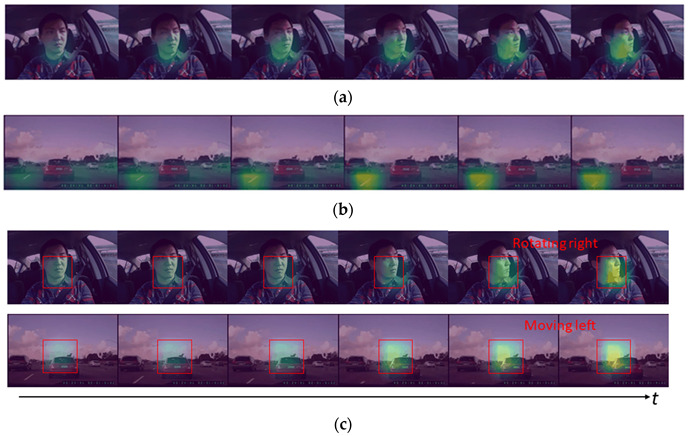
The focus area of input image sequences by the proposed STEDII-GRU in the scenario of a left-lane change. (**a**) I-only. (**b**) O-only. (**c**) In and out.

**Figure 9 ijerph-19-11819-f009:**
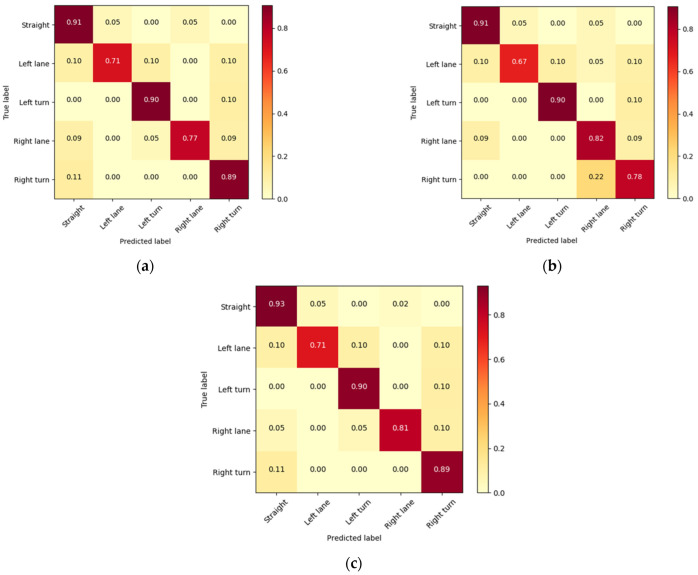
The confusion matrices for 5 driver intentions, based on the proposed models. The abscissa and the ordinate represent the probability of predicted and true labels. (**a**) STEDII-FC. (**b**) STEDII-FL. (**c**) STEDII-GRU.

**Figure 10 ijerph-19-11819-f010:**
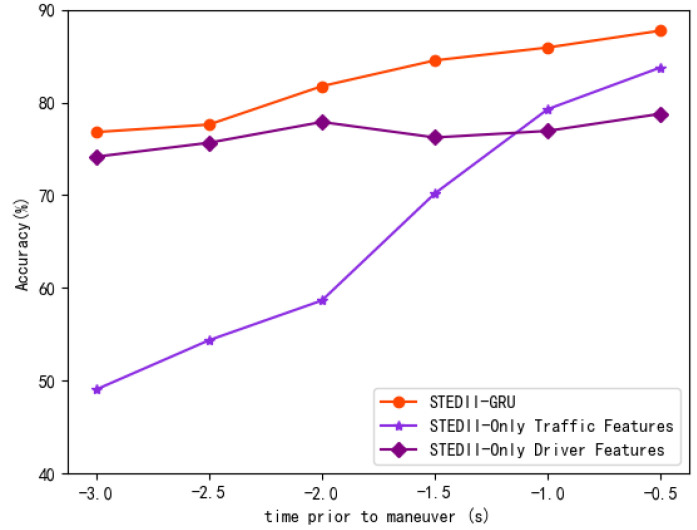
Prediction performance on different time-to-maneuver values for five intentions with the proposed models.

**Table 1 ijerph-19-11819-t001:** The various layers of SP.

Stage	Layer	Kernel	Stride	Number of Channels	Output Shape
	data layer	-	-	3	224 × 224
Stage 1	Conv3d	(1, 7, 7)	(1, 2, 2)	64	112 × 112
	BN3d	-	-	64	112 × 112
	Maxpool3d	(1, 3, 3)	(1, 2, 2)	64	56 × 56
Stage 2	conv3d_block1	(1, 3, 3)	(1, 1, 1)	[64, 64, 256]	56 × 56
	Identity_block × 2	(1, 3, 3)	-	[64, 64, 256]	56 × 56
Stage 3	conv3d_block1	(1, 3, 3)	(1, 2, 2)	[128, 128, 512]	28 × 28
	Identity_block × 3	(1, 3, 3)	-	[128, 128, 512]	28 × 28
Stage 4	conv3d_block2	(3, 3, 3)	(1, 2, 2)	[256, 256, 1024]	14 × 14
	Identity_block × 5	(3, 3, 3)	-	[256, 256, 1024]	14 × 14
Stage 5	conv3d_block2	(3, 3, 3)	(1, 2, 2)	[512, 512, 2048]	7 × 7
	Identity_block × 2	(3, 3, 3)	-	[512, 512, 2048]	7 × 7

**Table 2 ijerph-19-11819-t002:** The various layers of FP.

Stage	Layer	Kernel	Stride	Number of Channels	Output Shape
	data layer	-	-	3	224 × 224
Stage 1	Conv3d	(5, 7, 7)	(1, 2, 2)	8	112 × 112
	BN3d	-	-	8	112 × 112
	Maxpool3d	(1, 3, 3)	(1, 2, 2)	8	56 × 56
Stage 2	Conv3d_block2	(3, 3, 3)	(1, 1, 1)	[8, 8, 32]	56 × 56
	Identity_block × 2	(3, 3, 3)	-	[8, 8, 32]	56 × 56
Stage 3	Conv3d_block2	(3, 3, 3)	(1, 2, 2)	[16, 16, 64]	28 × 28
	Identity_block × 3	(3, 3, 3)	-	[16, 16, 64]	28 × 28
Stage 4	Conv3d_block2	(3, 3, 3)	(1, 2, 2)	[32, 32, 128]	14 × 14
	Identity_block × 5	(3, 3, 3)	-	[32, 32, 128]	14 × 14
Stage 5	Conv3d_block2	(3, 3, 3)	(1, 2, 2)	[64, 64, 256]	7 × 7
	Identity_block × 2	(3, 3, 3)	-	[64, 64, 256]	7 × 7

**Table 3 ijerph-19-11819-t003:** The comparison of different algorithms with different data inputs.

Algorithms	Data Source	*Acc* ± *SD* (%)	*F*1 *Score* ± *SD* (%)	*Param* (M)
3DResNet	I-only	83.1 ± 2.5	81.7 ± 2.6	85.26
	O-only	53.2 ± 0.5	43.4 ± 0.9	85.26
	In and out	75.5 ± 2.4	73.2 ± 2.2	170.52
ConvLSTM+3DResNet	I-only	77.40 ± 0.02	75.49 ± 0.02	46.22
	O-only	60.87 ± 0.01	66.38 ± 0.03	5.41
	In and out	83.98 ± 0.01	84.30 ± 0.01	57.92
TimeSformer	I-only	81.82 ± 0.48	80.51 ± 0.86	121.40
	O-only	60.81 ± 0.51	50.79 ± 1.38	121.40
	In and out	82.22 ± 0.66	78.24 ± 0.95	121.40
TANet	I-only	83.20 ± 1.91	82.84 ± 3.08	25.59
	O-only	64.14 ± 0.33	55.64 ± 0.42	25.59
	In and out	62.96 ± 0.52	55.10 ± 0.67	25.59
STEDII-GRU	I-only	81.69 ± 0.58	80.31 ± 1.86	34.82
	O-only	84.75 ± 0.59	85.30 ± 0.68	34.82
	In and out	92.06 ± 1.89	90.04 ± 2.22	34.82

“I-only” represents only the inside driver behavior videos were used. “O-only” represents only the outside traffic scene videos were used. “In and out” represents both inside and outside videos were used.

**Table 4 ijerph-19-11819-t004:** Comparison of the performance on each intention for different models.

Models	Lane-Keeping		Left-Lane Change		Left Turn		Right-Lane Change		Right Turn		*Acc* (%)
	Pr (%)	Re (%)	Pr (%)	Re (%)	Pr (%)	Re (%)	Pr (%)	Re (%)	Pr (%)	Re (%)	
SVM	83.43	66.93	68.24	50.37	55.32	67.69	73.54	52.48	23.45	87.31	65.85
HMM	61.70	74.35	80.00	66.67	58.33	46.66	64.00	69.56	72.72	67.34	67.74
Bi-LSTM	85.02	70.83	75.00	78.94	44.43	66.67	78.94	75.00	66.67	75.00	74.03
STEDII-FC	90.69	88.63	71.42	88.23	90.00	75.00	77.27	89.47	88.89	61.53	83.81
STEDII -FL	90.69	90.69	66.67	87.50	90.00	90.00	81.81	78.26	77.78	58.33	82.05
STEDII-GRU	93.02	88.89	71.42	88.23	90.00	75.00	77.27	94.45	88.89	61.53	84.92

## Data Availability

The Brain4Cars dataset that was analyzed in this study can be found at https://brain4cars.com/ (accessed on 2 April 2021). The kinetics-400 dataset presented in this study is openly available at [35,51].

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
