# Peer review of "Spatiotemporal Feature Enhancement Aids the Driving Intention Inference of Intelligent Vehicles"

_ijerph, 2022, doi:10.3390/ijerph191811819_

Round 1
Reviewer 1 Report
There are a few minor spelling and grammatical errors throughout the report. Otherwise, I thought the analyses good.
Author Response
Thank you for your careful review! We reconfirmed the description, and some incorrect writing has been rectified.
Reviewer 2 Report
The manuscript presents a practically interesting subject to study the driving intention inference of intelligent vehicles. The introduction contains appropriate citations. The methodology and analysis techniques appear to be sound. The results present several outcomes in well-organized fashion.
Some minor suggestions are as follows:
1. In the manuscript, the description should be double-checked and optimized.
2. All abbreviations must have a full spell when first appearing.
3. Driving intentions may also be influenced by the surrounding vehicles. Did the authors consider this issue?
Author Response
Thank you for your careful review! The responses are as follows.
1. In the manuscript, the description should be double-checked and optimized.
Response: We reconfirmed the description, and some incorrect writing has been rectified.
2. All abbreviations must have a full spell when first appearing.
Response: All abbreviations have a full spell in the revised manuscript.
3. Driving intentions may also be influenced by the surrounding vehicles. Did the authors consider this issue?
Response: We do take into account the impact of surrounding vehicles. In future work, we will collect more data to assist the trajectory prediction of vehicles, and the interaction between surrounding vehicles will be emphasized.
Reviewer 3 Report
This manuscript focuses on spatial-temporal feature enhancement to aid driving intention inference for an intelligent vehicle. There are many issues in it.
1.In lines 171-172, Eq. (1) is said to possess the CONV3D operation. But Eq. (1) does not contain the operation.
2.In lines 173-175, the parameters "c", "k", and "s" about Eq. (1) are introduced. But, Eq. (1) does not contain the parameters.
3.In Eq. (2), "Conv_Block" is used as a parameter and it is also used as operation. This is not appropriate.
4.In Eq. (3), "Identity_Block" is used as a parameter and it is also used as operation. This is not appropriate.
5.In lines 188-189, the spatio-temporal kernel sizes of the stage 2 -stage 5 are said to be [(3x1x1), (3x3x3),(3x1x1)]. But in Table 1, the spatio-temporal kernel sizes of the stage 2 -stage 3 are [(1x1x1), (1x3x3),(1x1x1)].
6.In line 218, what is "N"?
7.For Eqs. (4)-(7), what is "H_0"?
8.In line 249, how to determine the values of the six parameters with "W"?
9.In lines 249-250, how to determine the values of the three parameters with "b"?
10.In line 250, "bias parameter" and "tanhrepresent" should be replaced by "bias parameters" and "tanh represent", respectively.
11.In Eq. 8, how to calculate the value e^z for a vector z?
12.In line 277 and Fig. 7, "F1-score" is used. In lines 290 and 293, "F1 score" is used. they are inconsistent.
12.There is not Eq. (9).
14.In Table 3, "SE" should be replaced by "SD".
15.Table 1, Table 3, and Fig. 8 are not in a single page.
16.In Table 4 and Figure 9, the 5 driver intents are insufficient. Front-left turn (\) and front-right turn (/) must be added. So, there are 7 driver intents. Then, the comparison of the performance on each intent for different models in Table 4 and the confusion matrices in Fig. 9 must be renewed.
In conclusion, 5 driver intents are too simple and 7 driver intents are needed. Therefore, major revision is necessary.
Author Response
Thank you for your careful review! The responses are as follows.
1.In lines 171-172, Eq. (1) is said to possess the CONV3D operation. But Eq. (1) does not contain the operation.
Response: It is rectified now. Please see the revised manuscript.
2.In lines 173-175, the parameters "c", "k", and "s" about Eq. (1) are introduced. But, Eq. (1) does not contain the parameters.
Response: It is rectified now. Please see the revised manuscript.
3.In Eq. (2), "Conv_Block" is used as a parameter and it is also used as operation. This is not appropriate.
Response: It is rectified now. Please see the revised manuscript.
4.In Eq. (3), "Identity_Block" is used as a parameter and it is also used as operation. This is not appropriate.
Response: It is rectified now. Please see the revised manuscript.
5.In lines 188-189, the spatio-temporal kernel sizes of the stage 2 -stage 5 are said to be [(3x1x1), (3x3x3),(3x1x1)]. But in Table 1, the spatio-temporal kernel sizes of the stage 2 -stage 3 are [(1x1x1), (1x3x3),(1x1x1)].
Response: [k1, k2, k3] are the kernel sizes of the three CONV3Ds in Conv_Block or Identity_Block, respectively, and the first dimension is the time dimension. In SP, the time dimension of the kernel in the first two stages is set to 1, in order to reduce the time resolution of the slow path. Some incorrect writing has been rectified. Please see the revised manuscript.
6.In line 218, what is "N"?
Response: "N" represents the number of internal video frames, as explained in the previous paragraph of Table1 (in Line 198).
7.For Eqs. (4)-(7), what is "H_0"?
Response: "H_0" is the initial hidden state, which is initialized to an all-zero matrix by default at the beginning of training.
8.In line 249, how to determine the values of the six parameters with "W"?
Response: The six parameters with "W" are all trainable parameters, which are initialized by the pre-training weight parameters at the beginning. These parameters will be continuously updated during the training process, and the final values will be obtained after the model training is completed.
9.In lines 249-250, how to determine the values of the three parameters with "b"?
Response: The three parameters with "b" are all trainable parameters, which are initialized by the pre-training weight parameters at the beginning. These parameters will be continuously updated during the training process, and the final values will be obtained after the model training is completed.
10.In line 250, "bias parameter" and "tanhrepresent" should be replaced by "bias parameters" and "tanh represent", respectively.
Response: It is rectified now. Please see the revised manuscript.
11.In Eq. 8, how to calculate the value e^z for a vector z?
Response:The vector z (z1, z2, z3, z4, z5) is the output of FC layer in Fig.5. When softmax is called, each element of vector z is first mapped by y = e^{x} function to get (e ^z1, e^z2, e^z3, e^z4, e^z5). Then each element value of the z vector becomes zj / (e^z1 + e^z2 + e^z3 + e^z4 + e^z5), and finally each value is distributed in [0, 1].
12.In line 277 and Fig. 7, "F1-score" is used. In lines 290 and 293, "F1 score" is used. they are inconsistent.
Response: It is rectified now. Please see the revised manuscript.
13.There is not Eq. (9).
Response: It is reordered now. Please see the revised manuscript.
14.In Table 3, "SE" should be replaced by "SD".
Response: It is rectified now. Please see the revised manuscript.
15.Table 1, Table 3, and Fig. 8 are not in a single page.
Response: The layout of manuscript has been rearranged.
16.In Table 4 and Figure 9, the 5 driver intents are insufficient. Front-left turn (\) and front-right turn (/) must be added. So, there are 7 driver intents. Then, the comparison of the performance on each intent for different models in Table 4 and the confusion matrices in Fig. 9 must be renewed.
Response: The public dataset, Brain4Cars, used in this study does not include scenes corresponding to front-left turn (\) and front-right turn (/), so the inference work for 7 intents cannot be achieved at present. The intention inferences of front-left turn and front-right turn are also important, so if there are more data available in future, we would conduct the relevant study. Thank you for the sincere sugguestions!
Round 2
Reviewer 3 Report
Five driver intents are too simple, so this revised manuscript should be rejected.
Author Response
In the last paragraph of the Discussion section, we make the following additions to the limitations of this study: In addition, the dataset used in this study contains general driving scenarios and does not involve some specific scenarios. If there are more data available in future, more driving intentions’ inference can be conducted.